# A high-throughput target-based screening approach for the identification and assessment of *Mycobacterium tuberculosis* mycothione reductase inhibitors

Natalia Smiejkowska,[1,2] Lauren Oorts,[1] Kevin Van Calster,[1] Linda De Vooght,[1] Rob Geens,[2] Henri-Philippe Mattelaer,[3] Koen Augustyns,[3] Sergei V. Strelkov,[4] Dirk Lamprecht,[5] Koen Temmerman,[5] Yann G.-J. Sterckx,[2] Davie Cappoen,[1] Paul Cos[1]

**ABSTRACT** The World Health Organization's goal to combat tuberculosis (TB) is hindered by the emergence of anti-microbial resistance, therefore necessitating the exploration of new drug targets. Multidrug regimens are indispensable in TB therapy as they provide synergetic bactericidal effects, shorten treatment duration, and reduce the risk of resistance development. The research within our European RespiriTB consortium explores *Mycobacterium tuberculosis* energy metabolism to identify new drug candidates that synergize with bedaquiline, with the aim of discovering more efficient combination drug regimens. In this study, we describe the development and validation of a luminescence-coupled, target-based assay for the identification of novel compounds inhibiting *Mycobacterium tuberculosis* mycothione reductase ($Mtr_{Mtb}$), an enzyme with a role in the protection against oxidative stress. Recombinant $Mtr_{Mtb}$ was employed for the development of a highly sensitive, robust high-throughput screening (HTS) assay by coupling enzyme activity to a bioluminescent readout. Its application in a semi-automated setting resulted in the screening of a diverse library of ~130,000 compounds, from which 19 hits were retained after an assessment of their potency, selectivity, and specificity. The selected hits formed two clusters and four fragment molecules, which were further evaluated in whole-cell and intracellular infection assays. The established HTS discovery pipeline offers an opportunity to deliver novel $Mtr_{Mtb}$ inhibitors and lays the foundation for future efforts in developing robust biochemical assays for the identification and triaging of inhibitors from high-throughput library screens.

**IMPORTANCE** The growing anti-microbial resistance poses a global public health threat, impeding progress toward eradicating tuberculosis. Despite decades of active research, there is still a dire need for the discovery of drugs with novel modes of action and exploration of combination drug regimens. Within the European RespiriTB consortium, we explore *Mycobacterium tuberculosis* energy metabolism to identify new drug candidates that synergize with bedaquiline, with the aim of discovering more efficient combination drug regimens. In this study, we present the development of a high-throughput screening pipeline that led to the identification of *M. tuberculosis* mycothione reductase inhibitors.

**KEYWORDS** high-throughput screening, mycothione reductase, tuberculosis, drug discovery

Address correspondence to Paul Cos, paul.cos@uantwerpen.be.

Yann G.-J. Sterckx, Davie Cappoen, and Paul Cos contributed equally to this article.

K.T. and D.L. are employees of Janssen Pharmaceutica. N.V., N.S., L.O., K.V.C., L.D.V., R.G., H.P.M., K.A., S.V.S., D.C., Y.G.-J.S., and P.C. declare no conflicts of interest.

See the funding table on p. 20.

Tuberculosis (TB) is an airborne communicable disease and has long been declared a global public health threat. The World Health Organization (WHO) estimates that a quarter of the world's population has been infected with *Mycobacterium tuberculosis*, the etiological agent of TB (1). Although only 5%–10% of infected individuals develop active

TB, in 2021, the disease was the second leading cause of death from an infectious agent after coronavirus disease 2019 (COVID-19), with 1.6 million deaths worldwide (1). Besides its mortality, TB is characterized by a high morbidity, which heavily impacts the quality of life and socio-economic development of the affected regions (2). Current treatment options are insufficient to lower the incidence of TB, which exceeds 10 million cases per year worldwide (1). A long-term, multidrug treatment and the frequent occurrence of adverse effects complicate patient compliance. This, in turn, leads to an alarming increase in drug resistance, which has become an emerging problem in TB treatment and disease control (3, 4). There is thus a dire need for the discovery of drugs with novel modes of action and exploration of new drug targets.

*M. tuberculosis* is an intracellular pathogen that mainly resides in host macrophages. Here, the bacterium is exposed to complex dynamic host responses, such as the release of reactive oxygen species (ROS) and reactive nitrogen species (RNS), respectively (5). Extensive research has shown that redox homeostasis is paramount for *M. tuberculosis* survival within the macrophage niche (6–8). A significant amount of interest has been directed toward mycothiol (MSH), a low-molecular weight thiol that plays a role in maintaining a reducing environment as a key antioxidant (9, 10), in RNS protection (11), and NO signaling (the latter was documented in *Streptomyces* spp.) (12). During oxidative stress, MSH acts as a ROS scavenger through its oxidation to mycothiol disulfide (MSSM). MSH pools are replenished by mycothione reductase (Mtr) (13), a homodimeric NADPH-dependent enzyme which catalyzes the recycling reaction of MSSM to MSH (14, 15), thereby contributing to the maintenance of redox homeostasis. Hence, Mtr is the functional equivalent of glutathione reductases found in higher organisms while being unique to actinomycetes just like mycothiol itself (16). Although the enzyme's essentiality for *M. tuberculosis* remains under debate (17–20), MSH's protective role (and thus the importance of the Mtr recycling reaction) is underlined by the observation that MSH-deficient bacteria show an increased sensitivity to oxidative stress *in vitro* (21, 22). Such findings and the unique character of Mtr justify its exploration as a potential target in TB drug development campaigns as it offers a possibility of identifying highly selective inhibitory compounds. Despite its potential, Mtr (and the reaction it catalyzes) remains underexplored due to notorious difficulties in (i) the recombinant protein production of mycobacterial proteins (23, 24) and (ii) MSH and MSSM synthesis (25–27). Significant progress was made by Hamilton et al., who developed an innovative assay in which a substrate of the Mtr enzymatic reaction is chemically recycled (28). The authors designed a surrogate substrate, an asymmetric mycothiol disulfide, which enabled the synthesis of large substrate quantities needed for the assay. Although this offers advantages for continuous enzymatic assays, the use of chemical recycling is challenging in high-throughput screening (HTS) applications. In addition, the study outcome was based on absorbance readout, which lacks sensitivity and is highly susceptible to compound interference, hence being unsuitable for a target-based inhibitor HTS.

Here, we present the development of a target-based HTS assay that facilitates the identification of *Mycobacterium tuberculosis* mycothione reductase (Mtr$_{Mtb}$) inhibitors. First, the paper reports on the recombinant production of Mtr$_{Mtb}$ in *Escherichia coli* and its subsequent purification, yielding ~8 mg/L of bacterial culture. The oligomeric state and functionality of the recombinant protein was confirmed through in-solution biophysical techniques and enzyme activity assays, respectively. Next, recombinant Mtr$_{Mtb}$ was employed for the development of a highly sensitive, robust HTS assay by coupling enzyme activity to a bioluminescent readout, which was first thoroughly validated in a laboratory environment and subsequently applied in a semi-automated setting. In addition, we demonstrate the design of a target-based HTS panel, including selectivity and specificity secondary assays, which led to the identification of 19 primary hits with specific activity against Mtr$_{Mtb}$ out of a diverse library of ~130,000 compounds. The prioritized hits form two clusters displaying potencies in the low micromolar range. Finally, the intracellular activity and cytotoxic profiles of these primary hits are thoroughly evaluated.

## RESULTS

### An engineered SUMO-fusion construct enables high-yield recombinant production of Mtr$_{Mtb}$ in *E. coli*

As target-based HTS campaigns require relatively large amounts of pure, high-quality target protein preparations, we first sought to identify suitable conditions for the recombinant production of Mtr$_{Mtb}$ in *E. coli*. To overcome typically encountered challenges, the protocol's design included two elements to improve protein solubility and proper folding: (i) the use of a pET-based vector pETRUK (29) toward overexpression of the target protein as a fusion with a SUMO tag and (ii) co-expression with GroCOEX

ES-GroEL chaperones (encoded on pGro7 plasmid). The procedure included overexpression and initial isolation of N-terminally SUMO-tagged Mtr$_{Mtb}$, followed by tag cleavage and further purification (Fig. S1).

The recombinant production of SUMO-Mtr$_{Mtb}$ was performed in *E. coli* Express T7 strain transformed with both the pETRUK-Mtr$_{Mtb}$ and pGro7 plasmids. Induction of expression resulted in clear production of the target protein (MM ~62 kDa) as demonstrated by SDS-PAGE and anti-SUMO Western blot (Fig. S1). The target protein was purified through cation exchange chromatography: a first peak eluting at 30 mM $(NH_4)_2SO_4$ mainly contained the GroEL chaperone [MM ~60 kDa, verified by liquid chromatography-mass spectrometry (LC-MS) analysis; Fig. S1], while SUMO-Mtr eluted at ~200 mM $(NH_4)_2SO_4$.

Mtr$_{Mtb}$ was subsequently obtained by SUMO protease treatment, which was probed by SDS-PAGE and anti-SUMO Western blot to confirm successful SUMO cleavage (Fig. S1). LC-MS analysis on the excised band confirmed that the cleavage product was indeed intact Mtr$_{Mtb}$ (sequence coverage of ~79%). This sample was then dialyzed and subjected to two ion exchange steps performed in tandem: (i) a cation exchange to capture cleaved SUMO (and non-cleaved SUMO-Mtr$_{Mtb}$, if any) and (ii) an anion exchange to retain Mtr$_{Mtb}$. Untagged Mtr$_{Mtb}$ eluted as a single peak from the anion exchange column at ~90 mM $(NH_4)_2SO_4$, following confirmation of successful separation from the SUMO tag by SDS-PAGE. A final gel filtration step further polished the sample, thereby resulting in highly pure protein preparation, evidenced by the observation of a single protein band at 50 kDa on SDS-PAGE (Fig. S1). This optimized approach yielded ~8-mg Mtr$_{Mtb}$/L of bacterial culture, thereby providing abundant material for target-based HTS endeavors.

### Recombinant Mtr$_{Mtb}$ is well folded and enzymatically active

The oligomeric state and functionality of recombinant Mtr$_{Mtb}$ were assessed through a combination of in-solution methods [analytical size-exclusion chromatography (A-SEC), dynamic light scattering (DLS), and circular dichroism (CD) spectroscopy] and enzyme activity assays, respectively.

A-SEC and DLS were employed to determine the enzyme's oligomeric state in-solution and overall sample homogeneity. Both techniques demonstrate that the apparent molecular mass (MM$_{app}$) and apparent hydrodynamic radius (R$_{h,app}$) correspond well to the theoretical values expected for a dimer with a quasi-globular shape (Fig. 1A and B). In addition, the preparation is highly monodisperse as evidenced by a highly symmetrical AGF peak and a low polydispersity index measured in DLS. Far-UV CD spectroscopy was performed to further examine proper folding of recombinant Mtr$_{Mtb}$. The spectrum is typical for a protein with a Rossmann fold containing both α-helices and β-sheets (Fig. 1C) (30, 31). Finally, enzymatic activity of recombinant Mtr$_{Mtb}$ was confirmed through mycothiol disulfide reduction reaction (Fig. 2). The surrogate substrate, asymmetric mycothiol disulfide {5-[benzyl 2(-N-acetyl-L-cysteinyl) amino-2-deoxy-α-D-glucopyranoside]-dithio-2-nitrobenzoate [BnMS-TNB]} described by Hamilton et al. (28) was incorporated into the assay due to limitations arising from natural substrate synthesis. Upon addition of NADPH, Mtr$_{Mtb}$ catalyzes the reduction of the BnMS-TNB to benzylated mycothiol and 5-thio-2-nitrobenzoic acid (TNB). Figure 2B presents the result of kinetic

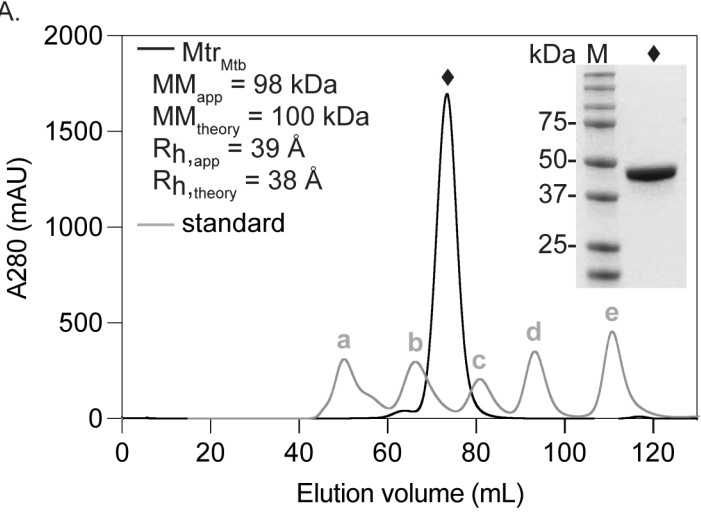

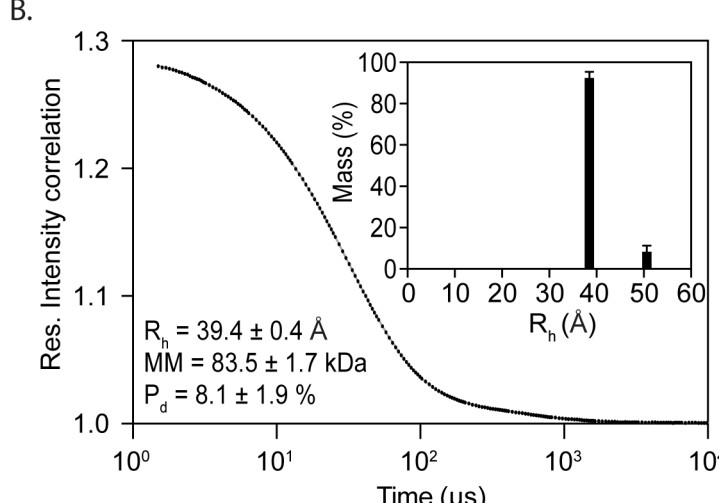

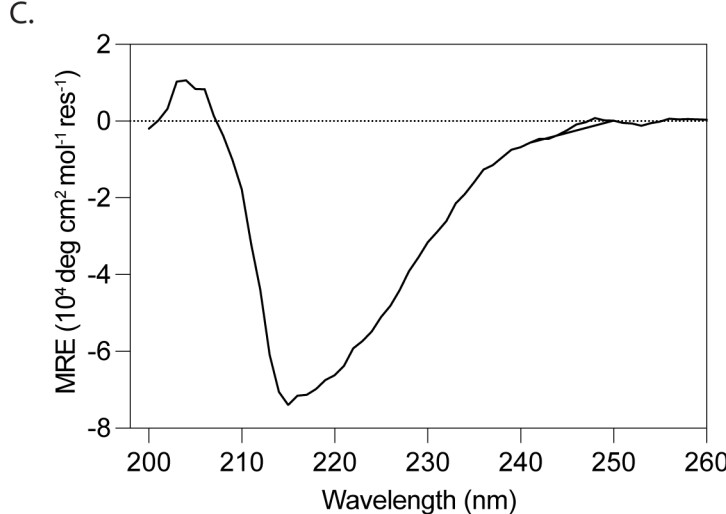

FIG 1   Quality control of recombinant $Mtr_{Mtb}$. (A) $Mtr_{Mtb}$ size exclusion chromatography profile. The experiment was performed on a Superdex S200 16/60 column. The black and gray traces represent the chromatograms of the $Mtr_{Mtb}$ and the gel filtration standard, respectively. The standard includes five proteins: (a) thyroglobulin [670 kDa, hydrodynamic radius ($R_h$) 85.8 Å]; (b) globulin (158 kDa, $R_h$ 51 Å);

**FIG 1 (Continued)**

(c) ovalbumin (44 kDa, $R_h$ 28 Å); (d) myoglobin (17 kDa, $R_h$ 19 Å); and (e) vitamin B12 (1,350 Da, $R_h$ 8.5 Å). The apparent molecular mass and hydrodynamic radius [apparent molecular mass ($MM_{app}$) and apparent hydrodynamic radius ($R_{h, app}$)] were derived based on the elution volumes of the standard proteins and compared to the theoretical values of the $Mtr_{Mtb}$ dimer obtained by analyzing the amino acid sequence ($MM_{theory}$ and $R_{h,theory}$). The inset shows an SDS-PAGE analysis of the $Mtr_{Mtb}$ elution peak indicated by a black diamond and pre-stained protein molecular weight marker in lane M. (B) $Mtr_{Mtb}$ DLS. Black dots and gray trace represent experimental and fitted data, respectively. The inset shows the $R_h$ distribution. MM, $R_h$, and polydispersity ($P_d$) were analyzed using Dynamics 7.1.9. (C) CD spectrum of $Mtr_{Mtb}$.

monitoring of TNB formation at 412 nm, proving a measure for $Mtr_{Mtb}$ enzymatic activity. Given the absence of reference inhibitors, a catalytically inactive mutant was designed to serve as a negative control. In this mutant ($Mtr_{Mtb}^{C39SC44S}$), both active site cysteines are mutated to serine (Cys39Ser/Cys44Ser), thereby halting electron transfer from FAD and subsequent mycothiol disulfide reduction. Protein production and quality control parameters were identical to the wild-type $Mtr_{Mtb}$ (Fig. S2), indicating that catalytic inactivation can be ascribed to the introduced point mutations and not misfolding. In the enzymatic assay performed according to the same protocol, no activity could be detected for $Mtr_{Mtb}^{C39SC44S}$ (Fig. 2C). Together, these results show that purified $Mtr_{Mtb}$ is well folded and enzymatically active and can thus be employed for the HTS assay development.

## Coupling enzyme activity to a bioluminescent readout enables HTS assay development

Due to its low sensitivity and high susceptibility to compound interference at wavelengths detecting NADPH depletion or TNB formation (340 and 412 nm, respectively), the absorbance readout was adapted for HTS by coupling the $Mtr_{Mtb}$ enzymatic reaction to a bioluminescent readout using NADP/H-Glo assay (Fig. 3A). Three critical points were identified during the optimization phase to guarantee a stable, robust, and reproducible assay: (i) quenching of the primary $Mtr_{Mtb}$ reaction (quenching point), (ii) determination of the substrate turnover of the primary $Mtr_{Mtb}$ reaction, and (iii) endpoint of the luminescent readout (readout time).

Since the assay detects both NADPH and $NADP^+$, the differential stability properties of these dinucleotides under low pH were exploited to quench the primary reaction. Addition of 1-M HCl degrades NADPH, therewith terminating the primary reaction without influencing the formed $NADP^+$ (32). To determine the linear range of the Mtr enzymatic reaction and to assess the amount of formed product, a series of reactions was carried out, quenched at 10-min intervals (Fig. 3B), followed by monitoring the emitted luminescent signal over time. Kinetic measurement of the luminescent signal allowed selection of three readout points from the linear reaction phase and subsequent quantification of the generated $NADP^+$ by comparison with the $NADP^+$ standard curve. In view of assay robustness, the $Z'$-factor and signal-to-background (S:B) ratio were determined for all quenching points at three readout times (Fig. 3C). Based on the progress curves, the quenching of the primary reaction was set to 40 min, thus not exceeding 15% of the substrate turnover to maintain the measurement in the initial reaction phase. To ensure the assay window, while taking into consideration limitations imposed by the instrument time, the readout point was established at 20 min. Following optimized reaction conditions (indicated by the combination of R1 and Q4 in Fig. 3C), the values for $Z'$-factor and S:B ratio were 0.67 and 14.4, respectively.

The selected quenching and readout points not only dictate the assay progress but also influence the screening throughput. Accordingly, sixty 384-well plates can be processed per batch by one operator. To meet HTS format requirements, the assay was miniaturized to a small-volume 384-well format. Due to the technical limitations arising from liquid dispensing systems, the readout reagent volume was reduced from 8 to 6 µL while maintaining the assay window (Fig. 4). Assay parameters are summarized in Table

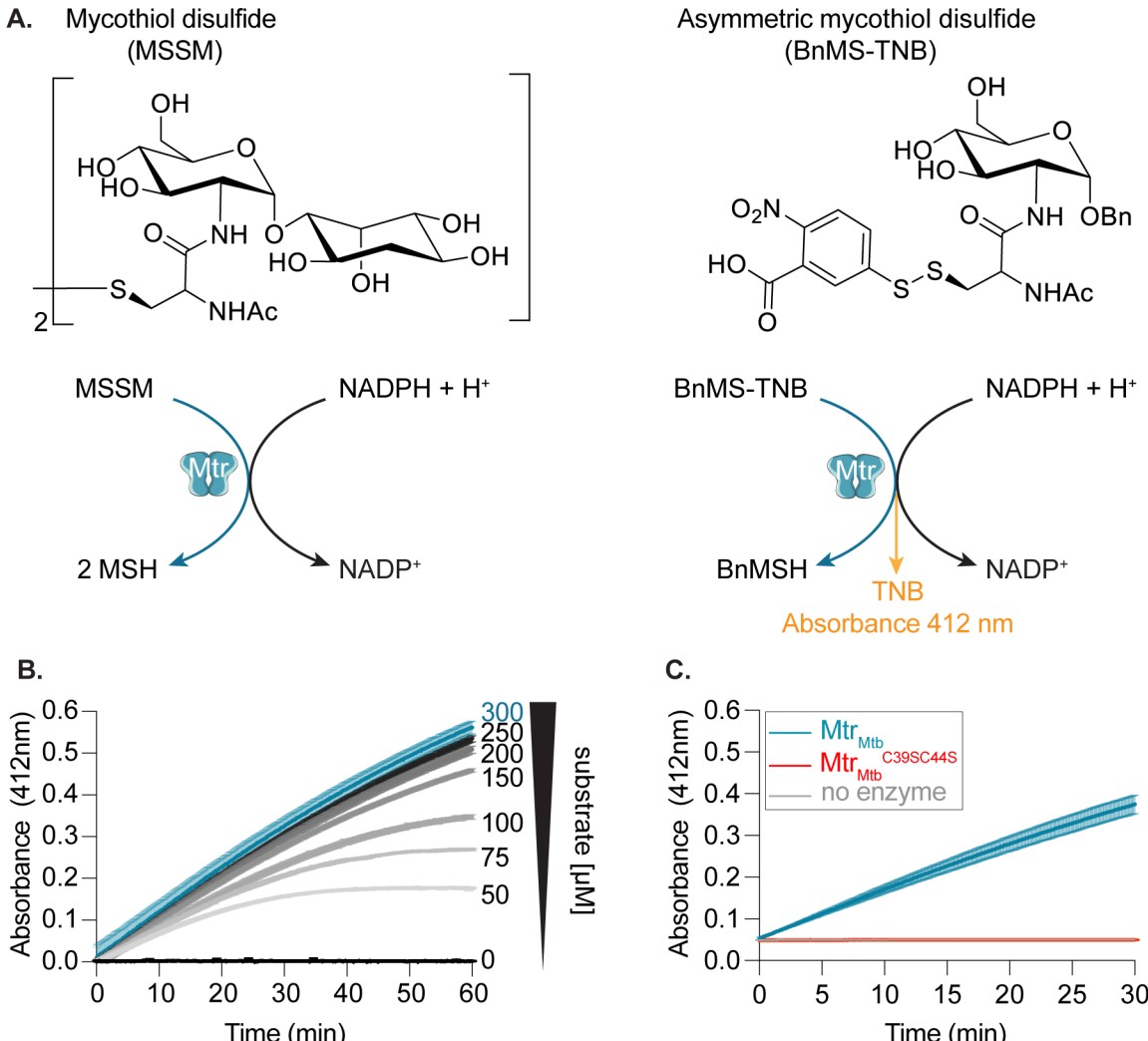

**FIG 2** Enzymatic activity of recombinant Mtr$_{Mtb}$. (A) Schematic representation of the Mtr reaction incorporating the natural reaction substrate mycothiol disulfide (MSSM, left) and the substrate analog employed for the development of the enzymatic assay (right). MSSM is reduced to mycothiol (MSH) by Mtr in an NADPH-dependent reaction. Due to the low availability and strenuous synthesis of the natural substrate, the catalytic activity of purified enzymes was estimated in the reduction reaction of asymmetric mycothiol disulfide (BnMS-TNB) as previously described (28). Upon addition of NADPH, Mtr reduces the substrate analog leading to the formation of TNB and benzylated mycothiol (BnMSH). TNB release is observed by an increase in absorbance at 412 nm. Reactions were performed in triplicate (duplicate for the no-enzyme control) in 50-mM HEPES, 50-mM NaCl, 0.05% bovine serum albumin (BSA), 0.01% Tween 20, and pH 7.5. (B) Optimization of the substrate concentration. Reaction conditions include 5-nM Mtr$_{Mtb}$, 150-µM NADPH, and varying substrate concentrations in 50-mM HEPES, 50-mM NaCl, 0.05% BSA, 0.01% Tween 20, and pH 7.5. The trace highlighted in blue represents the substrate concentration chosen for further studies. (C) Activity determination of Mtr$_{Mtb}$ and Mtr$_{Mtb}$$^{C39SC44S}$. The blue, red, and gray traces represent the reaction progress of the Mtr$_{Mtb}$, Mtr$_{Mtb}$$^{C39SC44S}$, and the no enzyme control, respectively. Reaction conditions include 150-µM NADPH, 300-µM substrate, and 5-nM Mtr$_{Mtb}$. Data are presented as average values ± standard deviation.

S1. The developed end-point assay was therefore validated for identification of Mtr$_{Mtb}$ inhibitors on HTS scale.

## HTS assay application on a diverse library leads to the identification of selective and specific Mtr$_{Mtb}$ inhibitors

To identify Mtr$_{Mtb}$ inhibitors, a diverse library of 137,585 chemicals was screened at a single 20-µM concentration per compound. A schematic overview of the HTS assay is presented in Fig. 5A. Throughout the procedure, assay robustness was confirmed as advocated by $Z'$-factor and S:B values higher than 0.6 and 4.0, respectively (Fig. 5B). The activity threshold for primary hits was set at a minimum of 25% normalized

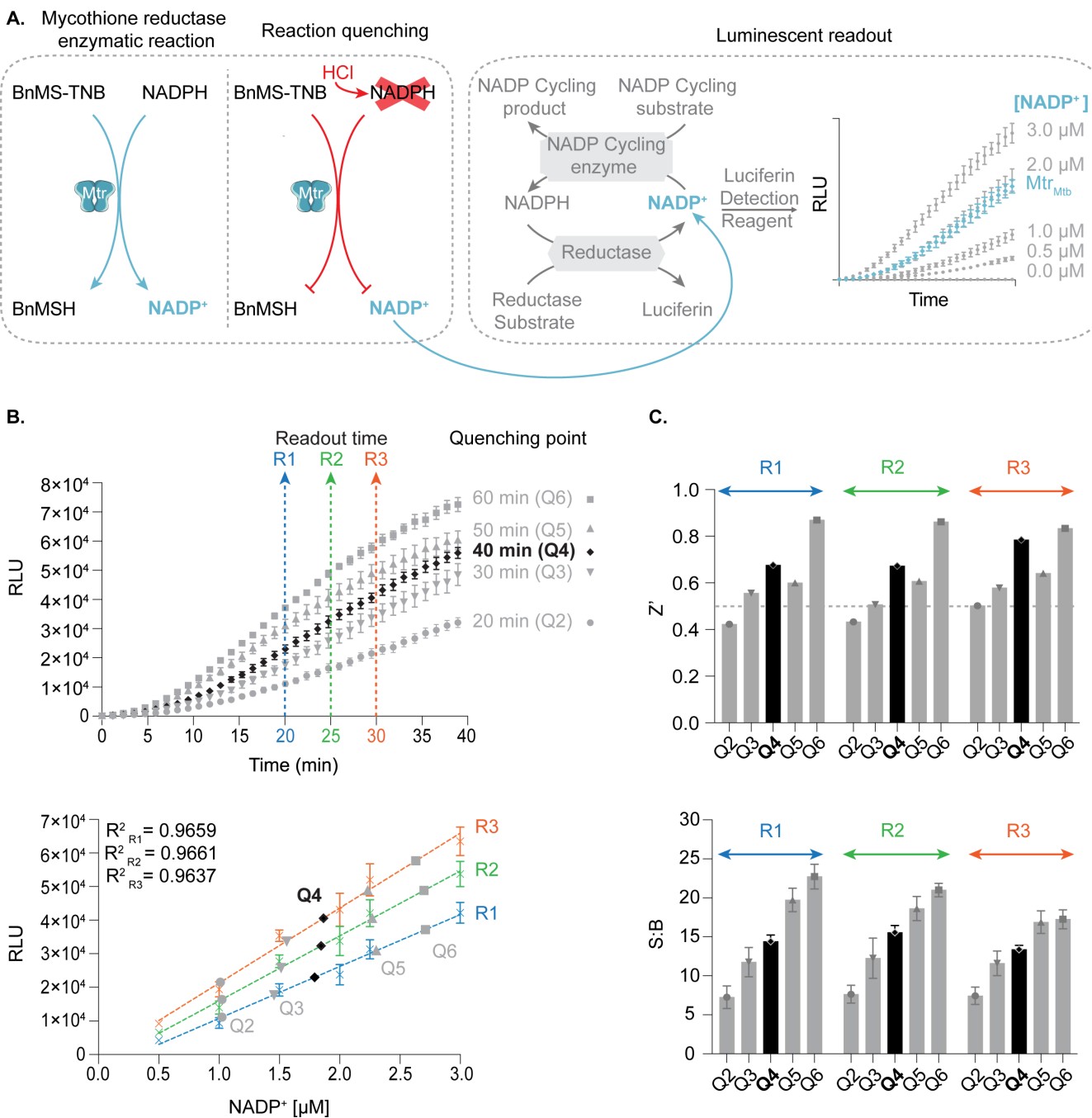

**FIG 3** Development of the bioluminescent coupled enzymatic assay. (A) Overview of the assay setup. The reaction is initiated by adding NADPH to the wells containing the assay substrate and $Mtr_{Mtb}$. During the reaction course, treatment with 0.8-M HCl disintegrates NADPH, thereby quenching the enzymatic reaction; nevertheless, the formed $NADP^+$ remains unaffected. After neutralizing the reaction by 1-M Tris, pH 8.0, NADP/H-Glo Assay (Promega) is added as a one-step mixture to the assay wells. The $NADP^+$ formed in the primary $Mtr_{Mtb}$ reaction is converted to NADPH by $NADP^+$ cycling enzyme. In the presence of NADPH, the proluciferin reductase substrate is reduced to luciferin by reductase. The detection of luciferin by Ultra-Glo rLuciferase is measured over time. A standard curve of $NADP^+$ served for the quantification of the formed $NADP^+$. (B) Assay optimization. Three critical steps were explored: quenching time point of the primary reaction, the readout time of the luminescence-coupled assay, and the substrate turnover. In the upper panel, the primary $Mtr_{Mtb}$ reaction was quenched at 10-min intervals (Q2, 20 min; Q3, 30 min; Q4, 40 min; Q5, 50 min; Q6, 60 min) followed by monitoring of the luminescent signal over time. Three readout time points were selected from the linear reaction phase (R1, 20 min; R2, 25 min; R3, 30 min) to analyze the substrate turnover. In the lower panel, the quenching points (Q2–Q6) at the three readout time points (R1–R3) were interpolated to the $NADP^+$ standard curve. Varying concentrations (0– 3 µM) of purified $NADP^+$ were included in the assay setup, omitting the addition of NADPH, enzyme, and assay substrate. The $NADP^+$ standard curve was obtained

**FIG 3** (Continued)

by plotting average net luminescence values [relative luminescence unit (RLU) of the signal − RLU of the background] for each readout time point (R1–R3) and performing linear regression analysis. The net luminescence values of the assay samples (Q2–Q6) were interpolated to the values in the standard curve to quantify the amount of formed NADP⁺ in the reactions. (C) Assay performance. The Z′-factor (upper panel) and the signal-to-background (S:B) ratio (lower panel) was calculated from measurements for each analyzed condition. Each point represents the average of triplicate measurements ± standard deviation. Black denotes selected conditions.

inhibition while exceeding the assay plate's standard deviation value threefold. As a result, 1,042 compounds were selected, representing a hit rate of 0.76%, which falls in the standard range for HTS (0.5%–1.0%). Frequency distribution of compound activities demonstrates Gaussian distribution centered at approximately 0% of inhibition shown in Fig. 5B. In the same figure panel, an examination of the heat map representing the distribution of activities across assay plates reveals a random activity distribution, suggesting no distinct systematic errors. By contrast, higher hit counts clustered in column 1 are presumably a result of liquid handling errors and were reassessed in the confirmation and counter screens. The potency of all identified hits was then determined in a dose-response fashion (confirmation screen), which also ruled out all false positives arising from technical errors. The activity was confirmed for 66% of initial compounds with the potencies ranging from 0.04 to 40.0 µM. Despite the clear advantages of HTS, rigorous assessment must be performed to challenge the high number of false-positive hits. Therefore, a counter screen was performed in which enzyme and substrate solutions were substituted by NADP⁺ such to exclude hits displaying activity for the assay format or components of the readout coupling system. Compounds inactive in the counter screen accounted for 57% of the initial hit number. The correlation of the potencies obtained in confirmatory and counter screens is presented in Fig. 5C. Based on the potency, Hill slope value, and dose-response curves, the 60 most promising entities were selected out of this subpanel. Further purity assessment of identified hit compounds resulted in 48% of tested compounds demonstrating purity greater than or equal to 80%. Owing to low purity, 10 compounds were excluded from the hit selection.

In the next step, the aim was to examine the primary hit specificity for the target protein. Given that Mtr$_{Mtb}$ serves as the functional equivalent of glutathione reductase (GR), the cross-activity of the potential leads was therefore determined. A commercially available glutathione reductase kit was adapted to a high-throughput scale. The linear range of the reaction was again determined, and the absorbance readout time was set to 10 min based on the TNB standard curve (Fig. S3A). Dose-response curve analysis

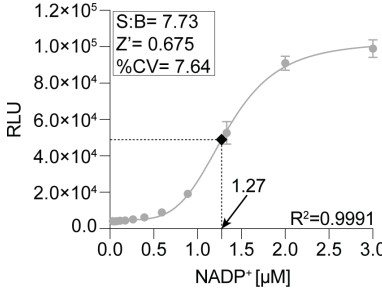

**FIG 4** Validation of the miniaturized assay. Optimized reaction conditions, including 100-pM Mtr$_{Mtb}$, 15-µM NADPH, and 30-µM assay substrate in 50-mM HEPES, 50-mM NaCl, 0.05% BSA, 0.01% Tween 20, and pH 7.5, were assessed on a 384-well small-volume plate (*n* = 16). Dilution (1.5-fold) of purified NADP⁺ (concentration range: 0–3 µM, *n* = 16) was included in the assay setup. Net luminescence was calculated for assay samples and NADP⁺ standard [relative luminescence unit (RLU) of the signal − RLU of the background]. The standard curve was fit by non-linear regression analysis. Interpolation of average net luminescence to the NADP⁺ standard curve allowed quantification of the formed NADP⁺ for the determination of the substrate turnover. Assay performance was determined by establishing the Z′, S:B, and percentage of coefficient of variation (% CV).

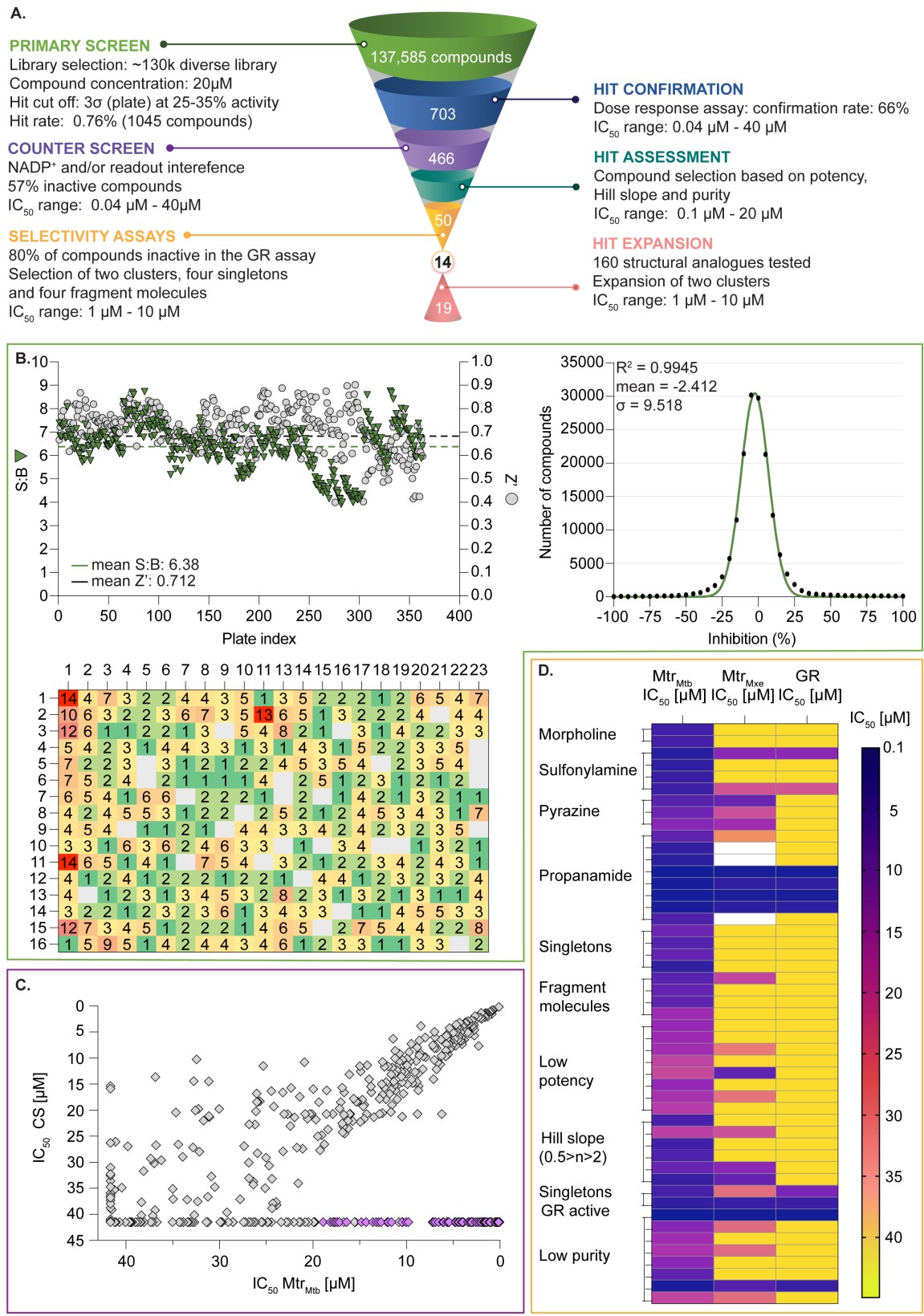

**FIG 5** Identification and characterization of Mtr$_{Mtb}$ inhibitors in high-throughput screening. (A) HTS workflow, in which each step has been color-coded: primary screen (green), hit confirmation (blue), counter screen (purple), hit assessment (turquoise), selectivity assays (yellow), and hit expansion (pink). (B–D) Results from the primary screen, counter screen, and selectivity assay, respectively, surrounded by boxes that are highlighted according to the above-mentioned color code. (Continued on next page)

**FIG 5 (Continued)**

(B) Assessment of the primary screen by determination of the plate performance as S:B ratio and $Z'$-factor (top left panel). Plates with a $Z'$-factor lower than 0.5 were excluded from the analysis. Frequency distribution of inhibition potency of all tested compounds (right panel). The compounds were distributed into bins with a width of 5 (black dots). A Gaussian curve (shown as a green trace) was fitted to describe the mean, standard deviation ($\sigma$) and the goodness of fit ($R^2$). Hit map (bottom left panel). The hit count distribution is depicted per well location. Positions marked in gray indicate zero identified hits. The higher number of hits in column 1 (positions indicated in red) likely results from dispensing errors. The potencies of all primary hits were reassessed in the hit confirmation. Control wells in columns 12 and 24 were not included in the analysis. (C) Correlation plot of average potencies of compounds tested in the confirmatory ($x$-axis) and counter ($y$-axis) screens. Compounds selected for further studies are highlighted in purple. (D) Potency heat map of compounds tested in the $Mtr_{Mtb}$ hit confirmation, glutathione reductase (GR) assay, and *Mycobacterium xenopi* Mtr ($Mtr_{Mxe}$) assay. The depicted compounds are color-coded according to their respective activity. White cells indicate that the activity was not determined.

revealed that, out of 50 available compounds tested, 80% were inactive in the assay herewith showing high specificity for the target protein. In parallel with the glutathione reductase assay, the selectivity of the hits was examined by testing their potential to inhibit the activity of *Mycobacterium xenopi* Mtr ($Mtr_{Mxe}$), which displays 83% sequence identity with $Mtr_{Mtb}$. A comparison of potencies presented as a heat map in Fig. 5D displays that all compounds showing activity in the GR assay are also active against $Mtr_{Mxe}$. Moreover, the compounds from the pyrazin cluster revealed potency against $Mtr_{Mxe}$; therefore, they were removed from further studies. Based on the selectivity and specificity assays, two families (morpholines and sulfonylamines, containing two and four members, respectively), four fragment molecules, and four singletons with a potency range 1.3–7.0 µM were prioritized. Two members of the sulfonylamine series show weak activity in the GR assay; however, based on the dose-response curve analysis, showing a flat curve at low activity range, these compounds were included in further studies. A hit expansion study was launched for the two prioritized series and singletons (excluding fragment molecules), where compounds with analogous structures were extracted from existing libraries at Janssen Pharmaceutica. The 160 selected molecules appeared to be asymmetrically distributed, with most molecules being an analog of one singleton. The developed screening panel of primary and secondary assays was employed to evaluate compound potency and selectivity. The morpholine and sulfonylamine series were expanded with eight and one compounds, respectively. Additionally, 15 inactive morpholine-like compounds were identified, out of which 8 can be classified as distant analogs. No activity was observed for tested singleton analogs, thereby excluding them from the study. Taken together, the HTS campaign followed by an established pipeline of secondary assays yielded 15 compounds clustered into two series: morpholines and sulfonylamines, containing 10 and 5 compounds, respectively, with micromolar potencies (1.3–6.0 µM) (Table 1). Additionally, four fragment molecules were triaged for further studies in whole-cell and intracellular assessment, leading to a total of 19 primary hits.

## The morpholine compound series displays non-specific activity in *M. tuberculosis*

Following the identification of selective $Mtr_{Mtb}$ inhibitors, their anti-mycobacterial activity was assessed in a whole-cell assay against *Mtb* H37Ra$^{lux}$ (in which the prokaryotic luciferase *luxab* genes were introduced). The use of H37Ra$^{lux}$ instead of the more virulent H37Rv strain offers two main advantages: (i) reduced experimental costs and the ability to work under L2 conditions (while exhibiting comparable minimal inhibitory concentrations (MICs) to most anti-tubercular drugs in rich medium (33) and (ii) the ability to measure a luminescent signal that is directly proportional to the metabolic activity of bacterial cells, guaranteeing a highly sensitive and easily scalable readout. Table 1 summarizes the experimental outcome. An analysis of the *Mtb* H37Ra$^{lux}$ MIC$_{50}$ values reveals the activity of four compounds from the morpholine series (Resipiri-1078, Resipiri-1079, Resipiri-1082, and Resipiri-1093). No inhibitory potency could be observed for other prioritized molecules. Furthermore, we tested the potency of compounds to

**TABLE 1** *In vitro* and *in cellulo* activity of prioritized hits

| | | | | | RAW264.7 | |
| Morpholine series | | | | | | |
| Compound | R | Y | $qIC_{50}^{a}$ (Mtr$_{Mtb}$) | $MIC_{50}$ (*Mtb* H37Ra$^{lux}$) | $MIC_{50}$ (*Mtb* H37Ra$^{lux}$) | $CC_{50}^{b}$ |
|---|---|---|---|---|---|---|
| Respiri-1079[d] | | N | 1.99 | 5.70 (±0.40) | 12.17 (±5.78) | 11.0 (±1.26) |
| Respiri-1082 | | N | 3.03 | 4.02 (±0.93) | 7.16 (±1.95) | 9.69 (±1.41) |
| Respiri-1078[d] | | N | 4.47 | 11.86 (±3.10) | 14.07 (±6.15) | 13.72 (±0.91) |
| Respiri-1093[d] | | N | 6.22 | 9.94 (±0.48) | >25 | >25 |
| Respiri-1084 | | CH | 2.88 | >25 | 10.22 (±2.32) | 11.92 (±0.52) |
| Respiri-1083[d] | | N | 2.34 | >25 | >25 | >25 |
| Respiri-1081[d] | | N | 3.75 | >25 | >25 | >25 |
| Respiri-1080[d] | | N | 5.98 | >25 | >25 | >5 |
| Respiri-1076[d] | | N | 41.65 | >25 | >25 | >25 |
| Respiri-1077[d] | | N | 41.65 | >25 | >25 | >25 |

| | | | | RAW264.7 | |
| Sulfonylamine series | | | | | |
| Compound | R | $IC_{50}$ (µM) Mtr$_{Mtb}$ | $MIC_{50}$ (µM) *Mtb* H37Ra$^{lux}$ | $MIC_{50}$ (µM) (*Mtb* H37Ra$^{lux}$) | $CC_{50}$ (µM) |
|---|---|---|---|---|---|
| Respiri-1179 | | 1.13 | >25 | >25 | >50 |

*(Continued on next page)*

eliminate intracellular bacteria, which is of utmost importance for the development of new TB treatments. Hence, we examined the intracellular activity of identified

**TABLE 1** *In vitro* and *in cellulo* activity of prioritized hits *(Continued)*

| Sulfonylamine series | | | | RAW264.7 | |
|---|---|---|---|---|---|
| Compound | R | IC$_{50}$ (µM) Mtr$_{Mtb}$ | MIC$_{50}$ (µM) *Mtb* H37Ra$^{lux}$ | MIC$_{50}$ (µM) (*Mtb* H37Ra$^{lux}$) | CC$_{50}$ (µM) |
| Respiri-1180 | N(ethyl)(ethyl) | 2.38 | >25 | >5 | >50 |
| Respiri-1181 | HN-butyl | 1.66 | >25 | > 25 | >50 |
| Respiri-1031 | HN-ethyl | 2.16 | >25 | >25 | >0 |
| Respiri-1075$^d$ | HN-methyl | 41.65 | >25 | >25 | ND |

| Fragment molecules | | | | RAW264.7 | |
|---|---|---|---|---|---|
| Compound | Chemical structure | IC$_{50}$ (µM) (Mtr$_{Mtb}$) | MIC$_{50}$ (µM) (*Mtb* H37Ra$^{lux}$) | MIC$_{50}$ (µM) (*Mtb* H37Ra$^{lux}$) | CC$_{50}$ (µM) |
| Respiri-1209 | | 5.63 | >25 | >25 | >50 |
| Respiri-1211 | | 12.97 | >25 | >25 | >50 |
| Respiri-1210 | | 6.11 | >25 | >25 | >50 |
| Respiri-1208 | | 4.83 | >25 | ND | 15.16 (±0.28) |

| Reference compounds | | | RAW264.7 | |
|---|---|---|---|---|
| Compound | IC$_{50}$ (µM) (Mtr$_{Mtb}$) | MIC$_{50}$ (µM) (*Mtb* H37Ra$^{lux}$) | MIC$_{50}$ (µM) (*Mtb* H37Ra$^{lux}$) | CC$_{50}$ (µM) |
| Bedaquiline | ND$^c$ | 0.01 (±0.001) | 0.07 (±0.29) | >50 |
| Tamoxifen | ND | ND | ND | 4.54 (±0.35) |

$^a$Average IC$_{50}$ value determined in the confirmation screen (dose-response assay).
$^b$Acute cellular toxicity, concentration of the test compound at which a reduction in cellular viability by 50% was measured. CC$_{50}$, 50% cytotoxicity concentration.
$^c$ND, not determined.
$^d$Compounds identified in the hit expansion study. All other test compounds originate from the screened library. All assays were performed in at least duplicate in two independent experiments.

hits in a macrophage infection assay in which RAW264.7 murine macrophages were infected with *Mtb* H37Ra$^{lux}$. Infected RAW264.7 cells were exposed to nine twofold serial dilutions of prioritized compounds. Intracellular activity was detected for four members of the morpholine family (Resipiri-1078, Resipiri-1079, Resipiri-1082, and Resipiri-1084); three among these also displayed activity in the whole-cell assay against *Mtb* H37Ra$^{lux}$ (Respiri-1078, Respiri-1079, and Respiri-1082). It is noteworthy that the range of intracellular MIC$_{50}$ (8–18 µM) of the morpholine series is comparable to the IC$_{50}$ determined in the enzymatic assay (1– 3 µM). Finally, the viability of RAW264.7

macrophages upon compound exposure was also investigated to evaluate the selectivity index of the identified hit compounds. Analysis of the dose-response curves revealed a decrease in cell survival within the same concentration range as in the intracellular activity assays, indicating inherent cytotoxicity. These findings indicate a clear need for structural studies and chemical improvements of the identified compounds (Table 1).

## DISCUSSION

The health threat posed by the surge in anti-microbial resistance calls for the discovery of new drug combinations and compounds with novel mechanisms. Multidrug regimens are indispensable in TB therapy as the combined use of two or more compounds acting on different molecular targets provides synergetic bactericidal effects, shortens treatment duration, and reduces the risk of resistance development. Currently, bedaquiline (BDQ) is the pillar of WHO treatment recommendations for multidrug-resistant TB. The discovery of BDQ (a diarylquinoline inhibiting ATP synthase) sparked the interest in targeting *M. tuberculosis* energy metabolism for drug development. Especially the combined action of BDQ with anti-microbial compounds targeting oxidative stress management appears to hold much promise. In a study by Lamprecht et al. (34), the combination of BDQ with ATP-depleting cytochrome $bc_1$ inhibitor (Q203) and ROS-generating clofazimine resulted in a strong correlation between bacterial killing and ROS production. Hence, the research focus within our European RespiriTB consortium lies on the exploration of *M. tuberculosis* energy metabolism to identify new drug candidates that synergize with BDQ to develop more efficient combination drug regimens. Given that ROS scavenging by MSH is paramount for maintaining mycobacterial redox homeostasis within the host environment, the enzymes involved in MSH (re)generation pathways represent attractive targets for the identification of novel drug combinations. Their exploration as potential TB drug targets is further justified by their unique character, which potentially offers a high probability of identifying highly selective inhibitory compounds. In this study, we report the development of a target-based HTS and a panel of secondary assays for the identification of novel compounds inhibiting *M. tuberculosis* Mtr, the enzyme that catalyzes the MSSM to MSH recycling reaction.

Our campaign started with the identification of suitable conditions for the recombinant production and purification of Mtr$_{Mtb}$ to obtain abundance of pure, high-quality protein preparations. Although *E. coli* is the standard bacterial expression host for recombinant protein production, overexpression of mycobacterial proteins is known to be cumbersome as it frequently results in the formation of inclusion bodies (35). Strategies to facilitate protein folding and enhance protein stability/solubility lie in the incorporation of polypeptide tags, co-expression of folding chaperones, and/or the use of expression hosts closer to the native organism (e.g., *Mycobacterium smegmatis*) (23, 24, 36–40). The protocol described in this paper enables recombinant production of Mtr$_{Mtb}$ in *E. coli* through the combined use of the cleavable SUMO-tag (29, 41) and co-expression of GroES-GroEL (42) chaperones. The production and purification strategy provides enhanced yields of pure, high-quality Mtr$_{Mtb}$ preparations to sustain target-based HTS endeavors.

While phenotypic screens are highly popular currently, target-based HTS offers the opportunity to explore new modes of action and novel chemotypes in the drug discovery portfolio. Advancements in technology allow for judicious target validation and profound characterization of biological pathways, thereby enhancing the likelihood of success of target-based HTS. Probing large compound libraries through HTS campaigns is commonly performed in 384- and 1,536-well plate formats, which require assay miniaturization and highly sensitive and robust readout systems. Currently described assays to screen for compounds that specifically inhibit reductases from human pathogens are based on absorbance readouts (43). While the latter are relatively straightforward and cheap, these are susceptible to compound interference thereby leading to numerous false-positive hits and low sensitivity in the detection of enzyme inhibition (44). To address these challenges, we coupled the Mtr reaction to

a bioluminescence readout that detects the amount of formed $NADP^+$. The result is a highly accurate, robust, and reproducible assay as evidenced by a mean plate $Z'$ value of 0.7, exceeding the excellent-quality assay threshold of 0.5 (45), and S:B of 6.38. Screening of a diverse library of ~130,000 compounds yielded a hit rate of 0.76%, followed by the development of a pipeline of secondary assays to assess hit reproducibility and detect assay artifacts. Among the initial hits, 66% reconfirmed activity in the 11-point dose-response assay, nevertheless 57% displayed direct interference with the coupled enzyme, $NADP^+$ or other assay components and were therefore excluded from further profiling. Through meticulous hit assessment performed at this stage, we selected 60 molecules primarily based on their potency and dose response curves. Furthermore, the distinct character of mycothiol in Actinomycetes was favorable in deciphering the hit selectivity, a highly desirable feature in target-based drug discovery. Knowing that glutathione is a human counterpart of mycothiol, we assessed the activity of the prioritized compounds in a GR assay, which revealed selectivity for $Mtr_{Mtb}$ of 76% of the tested molecules. Moreover, our investigation aimed to determine the potential cross-reactivity of primarily selected compounds with Mtr of other mycobacterial species. $Mtr_{Mxe}$, chosen for its relatively high sequence identity (83%) with $Mtr_{Mtb}$, yielded interesting results. Notably, all molecules inhibiting GR displayed activity in $Mtr_{Mxe}$, indicating their promiscuous character in targeting reductases. For this reason, the propanamide series was excluded from further consideration. The remaining selective and specific compounds were ranked, from which we selected 14 compounds with high purity, displaying low micromolar potencies and favorable kinetic profiles. These compounds formed two distinct clusters, four singletons, and four fragment molecules. To gain initial insights into the structure-activity relationship, we explored the compound libraries available at Janssen Pharmaceutica in search of related compounds. This led to the expansion of the morpholine series with eight additional molecules, hence creating a 10-compound series. Although the potency of the analogs lies in the same micromolar range, selective inhibition of Mtr by analogous molecules is particularly reassuring for the initial selection of the morpholine compound series. Nevertheless, the exhibition of cellular activity is highly unlikely for molecules that do not reach submicromolar potencies.

Given that a typical HTS library often contains more than one million compounds (10 times larger than the one chosen for this study), an HTS screen conducted on a larger scale than in this paper could potentially lead to the discovery of more potent scaffolds. Notwithstanding the relatively limited library size, this work demonstrates the successful assay development resulting in the identification of $Mtr_{Mtb}$ inhibitors selective over closely related reductases. Overall, we have identified two novel compounds series, displaying high target selectivity and specificity. While the $Mtr_{Mtb}$ inhibitors described herein show low potency, high-resolution structure determination of $Mtr_{Mtb}$ complexed with morpholines would reveal their (potentially novel) mode of action and offer a rational basis for series optimization. Although a low potency currently hampers further work on the morpholine series, the established HTS discovery pipeline delivered initial $Mtr_{Mtb}$ inhibitors and laid the foundation for future efforts in developing robust biochemical assays for the identification and triaging of inhibitors from high-throughput library screens.

## MATERIALS AND METHODS

### Generation of the expression plasmid

The DNA sequence of $Mtr_{Mtb}$ (Rv2855; UniProt accession ID: P9WHH3) was amplified by polymerase chain reaction from genomic DNA, isolated from *M. tuberculosis* H37Ra. The amplified gene was purified from agarose gel (New England Biolabs, cat. # T1020S). Next, the obtained amplicon was inserted into the pETRUK vector digested with HindIII-HF and KpnI-HF (New England Biolabs) using HIFI assembly (New England Biolabs). The pETRUK vector, a derivative of the previously described pETHSUL vector (29), is a SUMO-fusion

plasmid encoding MTR protein with an N-terminal SUMO-tag. Due to low efficiency of cleavage by SUMO-protease, a glycine residue was added between the C-terminal SUMO-tag and the first methionine residue of Mtr$_{Mtb}$, such that the purified protein contains an N-terminal glycine. The final construct was verified by Sanger Sequencing.

## Expression and purification of Mtr$_{Mtb}$

Mtr$_{Mtb}$ was recombinantly produced in *E. coli* T7 Express, transformed with two plasmids: (i) the pETRUK plasmid carrying the SUMO-tagged Mtr$_{Mtb}$ construct (Mtr$_{Mtb}$-pETRUK, *lac* promotor, ampicillin resistance) and (ii) the pGro7 plasmid encoding the GroEL-GroES chaperone proteins (*araB* promotor, chloramphenicol resistance; Takara Bio, cat. #3340). In a two-step transformation procedure, the pGro7 plasmid was first transformed into chemocompetent *E. coli* T7 Express cells (New England Biolabs, cat. # C2566H) by heat shock. Single colonies were selected on Luria-Bertani (LB) agar plates containing 10-µg /mL chloramphenicol. Next, transformants were cultured in LB media and chemocompetent cells were generated to heat-transform the Mtr$_{Mtb}$-pETRUK plasmid. Single colonies were selected on LB agar plates containing 100-µg /mL ampicillin and 10-µg /mL chloramphenicol. Pre-cultures of transformants were grown in LB media at 24°C with aeration (150 rpm). After overnight incubation, 10 mL of pre-cultures was inoculated into 2-L LB media with 100 µg/mL ampicillin, 10-µg/mL chloramphenicol, and 1-mL SE15-anti-foam and cultured at 24°C with aeration (150 rpm). At an OD$_{600}$ of 0.4, 0.5-mg/mL arabinose was added to induce the chaperone gene expression. The temperature was lowered to 18°C at an OD$_{600}$ of 0.6, followed by an induction of *mtr* expression at an OD$_{600}$ of 0.8 by addition of 0.2-mM isopropyl β-d-1-thiogalac-topyranoside (IPTG). Cultures were incubated overnight with aeration (150 rpm) and subsequently harvested by centrifugation (rotor 12269; Biosafe, Sigma, 30 min, 12,000 × *g*, 4°C). Cell pellets were resuspended in buffer A [50-mM HEPES, 50-mM (NH$_4$)$_2$SO$_4$, 5-mM β-mercaptoethanol, pH 7.5] supplemented with 10-mg/mL lysozyme and flash frozen at −80°C. Defrosted cell pellets were diluted 1:1 in lysis buffer with addition of 80 U of Cryonase Cold-Active Nuclease (Takara Bio, cat. 2670A) and 10-mM MgCl$_2$. Next, cells were lysed by sonication in three 1-min cycles (Sonics VCX-130; 1-s pulse and 1-s pause at 70% amplitude) with 20-min rest on ice in between cycles. The supernatant of the lysate was obtained through centrifugation (45 min, 16,000 × *g*, 4°C; rotor #12269; Sigma) and subsequently filtered with a low binding 0.45-µm filter (ThermoFisher, cat. #F2500-5) prior to loading on a HiTrap SP HP column (Cytiva, cat. #17115201) pre-equi-librated with buffer A at 2 mL/min. The column was then washed with five column volume of buffer A. SUMO-tagged Mtr$_{Mtb}$ was eluted at 2 mL/min by linear gradient of buffer B (50-mM HEPES, 1-M (NH$_4$)$_2$SO$_4$, 5 mM ß-mercaptoethanol, pH 7.5) over 30 column volumes. Fractions corresponding to the SUMO-tagged Mtr$_{Mtb}$ were collected and pooled, followed by a treatment with SUMO protease (41) for 30 min (in-house, 1-mg SUMO hydrolase cuts 250 mg of protein). The sample was then dialyzed overnight against 1-L buffer A at 4°C following a second dialysis step of 2 hours. To separate the protein mixture from the cleaved SUMO tag and SUMO hydrolase, a tandem cation-anion exchange was performed. The sample was loaded on the HiTrap SP HP column connected to a HiTrap Q HP (Cytiva, cat. #17115401) at 1 mL/min. After decoupling of the cation exchange column, the anion exchange column was washed with five column volumes of buffer A. Mtr$_{Mtb}$ was eluted from the anion exchange column by a linear gradient of buffer B over 30 column volumes at 2 mL/min. Collected fractions were concentrated to a final volume of 4 mL. The size exclusion chromatography (SEC) was performed on a HiLoad 16/60 Superdex 200-pg column (Cytiva, cat. #28989335) pre-equilibrated with 1.5 column volumes of buffer C (25-mM HEPES, 50-mM NaCl, 5-mM β-mercaptoethanol, pH 7.5). Mtr$_{Mtb}$ was eluted at 1 mL/min. Collected fractions were concentrated up to 5 mg/mL and stored at −80°C. The progress of the protein production and purification was monitored by SDS-PAGE.

The site-directed mutagenesis and cloning of the Mtr$_{Mtb}$ Cys39Ser/Cys44Ser double-mutant (Mtr$_{Mtb}$$^{C39SC44S}$) were outsourced to GenScript. The recombinant

productions and purifications of Mtr$_{Mtb}$$^{C39SC44S}$ and Mtr$_{Mxe}$ were performed as described above.

## Analytical SEC

Analytical SEC was performed on a HiLoad 16/60 Superdex 200-pg column (Cytiva, cat. #28989335) in buffer C. Samples of 3 mL of Mtr$_{Mtb}$ (at 7 mg/mL) and of the Bio-Rad gel filtration standard were injected and eluted at a flow rate of 1 mL/min. The apparent molecular mass and hydrodynamic radius of Mtr$_{Mtb}$ and Mtr$_{Mtb}$$^{C39SC44S}$ were determined according to as previously described (46).

## Circular dichroism spectroscopy

CD spectra of Mtr$_{Mtb}$ and Mtr$_{Mtb}$$^{C39SC44S}$ were recorded on a MOS-500 spectrometer (BioLogic). Scans were recorded over a wavelength range of 190–260 nm in steps of 1 nm, a bandwidth of 1.0 nm, and an acquisition time of 0.5 s. Spectra were obtained at room temperature (RT) at a concentration of 0.2 mg/mL in buffer C. The raw data were corrected for buffer sample following the calculation of the mean residue ellipticities (degrees cm$^2$/dmol × res) according to equation 1, with MM, n, C, and l representing the molar mass (Da), the number of residues, the concentration (mg/mL), and the path length (cm), respectively:

$$MRE = \frac{\theta \times MM}{n \times C \times l} \qquad (1)$$

## Dynamic light scattering

Dynamic light scattering data were collected on a DynaPro NanoStar instrument (Wyatt Technology) at 25℃. Samples of 50-µL Mtr$_{Mtb}$ and Mtr$_{Mtb}$$^{C39SC44S}$ at 0.2 mg/mL prepared in buffer C were filtered with Costar Spin-X 0.22-µm centrifuge filters (Corning) prior to data collection using disposable DLS cuvettes (Wyatt Technology). Data were processed and analyzed using Dynamics version 7.1.9.

## Enzymatic activity assay

Enzymatic activities of purified Mtr$_{Mtb}$ and Mtr$_{Mtb}$$^{C39SC44S}$ were determined by adapting a previously described assay (28). With the scarcity of the natural substrate (MSSM), the substrate analog, BnMS-TNB, was synthetized (supplemental information 1) and used as reaction substrate. The reaction was performed in buffer D (50-mM HEPES, 50-mM NaCl, 0.05% BSA, 0.01% Tween 20, pH 7.5) in transparent 384-well plates (Greiner Bio-one, cat. #781061). Mtr$_{Mtb}$ (5 nM) or Mtr$_{Mtb}$$^{C39SC44S}$ and varying substrate concentrations (0–300 µM) were added to the plate. Control wells without enzyme were included in the experimental setup. The reaction was initiated by addition of 150-µM NADPH (Roche, cat. #10107824001). An increase in absorbance at 412 nm corresponding to the formation of TNB was monitored kinetically over 60 min in the final reaction volume of 30 µL on Spectramax384 (Molecular Dimensions). Follow-up assays were carried out on 384-well plates containing 300-µM substrate, 150-µM NADPH and 5-nM enzyme in 30-µL final volume.

## Assay coupling to the end-point luminescent readout

The end-point assay was developed using NADPH-Glo Assay (Promega, cat. #G9082). The linear range of the kit was established by performing serial dilution of NADP$^+$. Due to the low NADP$^+$ detection limit, reaction conditions were optimized to 100-pM Mtr$_{Mtb}$, 30-µM substrate and 15-µM NADPH. To establish the linear velocity phase of the primary reaction, a series of reactions were stopped every 10 min from 20 to 60 min by addition of 1-M HCl. The NADP$^+$ standard curve was included in the plate setup to determine the reaction turnover rate. Following the manufacturer's protocol, the plate was incubated for 30 min at 25℃. Next, reaction mixtures were neutralized with 1.5-M

NaOH following 30-min equilibration. The plate was centrifuged for 1 min at $200 \times g$. The readout solution was added to the wells in a 1:1 ratio. Reaction progress was measured kinetically over 50 min on TECAN M200 (integration time 0.3 s). The time of the end-point reading was set based on the substrate turnover determined from the NADP$^+$ standard curve. The quality and robustness of the assay were examined by calculation of the Z-factor, according to equation 2 (45) and signal-to-background ratio (S:B; S:B = mean signal/mean background).

$$Z = 1 - \frac{3SD \ of \ sample + 3SD \ of \ control}{|mean \ of \ sample - mean \ of \ control|} \qquad (2)$$

## High-throughput enzymatic assay

### Primary screen

The assay was carried out on 384-well black, low-volume plates (Greiner, cat. #784076) at room temperature. All dispensing steps were performed using a MultiDrop combi reagent dispenser (ThermoFisher). Test compounds, dissolved in 100% dimethylsulfoxide (DMSO), were evaluated at 20 µM in the final reaction volume of 6 µL. Columns 12 and 24 on each test plate served as negative (no inhibitor) and positive (no enzyme/substrate) controls, respectively. The enzyme-substrate and NADPH mixtures were prepared in buffer D. The reaction was initiated by addition of NADPH (15 µM) to the wells containing an enzyme-substrate solution (100 pM and 30 µM, respectively) and immediate plate mix for 4 s, 2,000 rpm. Following a 40-min reaction course, the reaction was stopped by adding 0.8-M HCl. Plates were then equilibrated for 30 min. After adding the neutralizing solution (1-M Tris), plates were centrifuged for 1 min at $200 \times g$ followed by a 30-min incubation. The NADPH-Glo Assay kit was applied (6 µL), and the luminescence signal was measured at 20-min reaction time (0.05 s integration time) by an EnVision 2105 (PerkinElmer). Control plates containing 1.5-fold serial dilution of NADP$^+$ (Roche, cat. #10128031001) in reaction buffer, with a top concentration of 3 µM, were included in the experimental run omitting the enzyme-substrate and NADPH addition. In addition, six homogenous plates (without compounds) were added to each experiment to monitor the dispensing errors and plate effects: three at the beginning of the series, two in between reading batches, and one at the end of the plate series. Assay performance was determined by calculating Z′ factor, S:B ratio, and robust standard deviation. Recorded signal was normalized. Test compounds were considered primary hits when their normalized activity was higher than 25% and exceeded three times the plate's standard deviation.

### Secondary screens

In the confirmation assay, the dose-response plates were prepared by transferring 50 nL of compounds (in 100% DMSO) with an Echo 555 acoustic dispenser (Labcyte). The potency of primary hits was evaluated in an 11-point, one in two dilution series of test compounds with a top concentration of 40 µM, following the standard assay steps described above. Columns 12 and 24 on each test plate served as negative (no inhibitor) and positive (no enzyme/substrate) controls, respectively. The counter screen was performed in an analogous manner, substituting the components of the primary enzymatic reaction (enzyme, substrate, and NADPH) with 2-µM NADP$^+$. Negative (no inhibitor) and positive (no NADP$^+$) controls were included on each test plate in columns 12 and 24, respectively. For the selectivity assay incorporating Mtr$_{Mxe}$, reaction conditions were optimized to 5-nM Mtr$_{Mxe}$, 30-µM NADPH, and 60- µM substrate. All primary hits were tested in a dose-response manner according to the assay protocol described in section I and including standard controls. To define the specificity of the primary hits, the glutathione reductase assay (Abcam, cat. #ab83461) was adapted to the 384-well plate format. The potency of the primary hits to inhibit human glutathione reductase (Sigma-Aldrich, cat. #G9297) was determined in a reaction carried

out on transparent plates (Greiner, cat. #781061) containing 24-nM enzyme in the final volume of 40 µL. The dose-response plates were prepared by transferring 320 nL of compounds (in 100% DMSO) with an Echo 555 (Labcyte). Following a 10-min reaction course, absorbance at 405 nm was measured by EnVision 2105 (PerkinElmer). Columns 12 and 24 on each test plate served as negative (no inhibitor) and positive (no enzyme/substrate) controls, respectively. All secondary assays were performed in duplicate in two independent experiments. Data analysis was performed using Genedata Screener. Luminescent signal was normalized. Non-linear regression model served to generate dose-response curves and determine $IC_{50}$. Assay performance was established by calculating $Z'$ factor, S:B ratio, and robust standard deviation.

## Bacterial strains and cultures

*M. tuberculosis* H37Ra (ATCC 25117) harboring reporter plasmid pSMT1 encoding for *Vibrio harvei* luciferase (47) (H37Ra$^{lux}$) was grown in 7H9 media supplemented with 10% oleic albumin dextrose catalase (Becton-Dickinson, cat. #212351), 0.2% glycerol, 0.05% tyloxapol and 100-µg/mL Hygromycin B (Roche, cat.# 10843555001) at 37°C shaking (150 rpm). Equivalent growth media, supplemented with 10% oleic albumin dextrose catalase (Becton-Dickinson, cat. #212351), 0.2% glycerol, 0.05% tyloxapol, was used for culturing *M. xenopi* (DSM 43995) at 42°C shaking (150 rpm).

## Whole cell-based *in vitro* evaluation of inhibitors

The extracellular activity of identified inhibitors was determined in a dose-response fashion. Nine twofold serial dilutions with a top concentration of 25 µM were spotted on black opaque 96-well plates (Greiner Bio-One, cat. #655077). An inoculum size of $10^4$ relative luminescence unit H37Ra$^{lux}$ in 200 µL was added to each assay well. Columns 11 and 12 on each assay plate served as high (no growth inhibition) and low (100% growth inhibition) controls, respectively. To control evaporation and minimize the well-to-well variations, the outer wells of each plate were filled with sterile water and plates were kept in plastic bags inside a 37°C and 5% $CO_2$ incubator. The luminescence signal was measured at day 7 by injecting 1% decanal, which serves as the luciferase substrate, into each well (25 nL) using Glomax Discover microplate reader (Promega). The relative luminescence units were normalized to percentages of cell survival. The dose-response curves and the MICs were generated in GraphPad Prism version 9 by a non-linear regression model.

## Intracellular activity of the inhibitors

All media and supplements for cell culturing were purchased at ThermoFisher Scientific. The RAW264.7 murine macrophage-like cell line (ATCC, TIB-71) was cultured in Dulbecco's Modified Eagle Medium (DMEM; Gibco, cat. #41965–039) supplemented with 10% fetal bovine serum (FBS; Gibco, cat. #10270106), 2-mM L-glutamine, 100-U/mL penicillin-strep-tomycin (Gibco, cat. #15140122) at 37°C and 5% $CO_2$. An *M. tuberculosis* H37Ra$^{lux}$ culture ($OD_{600}$ 0.6–0.8) was washed twice with Dulbecco's phosphate buffered saline (DPBS) and resuspended in a basal uptake buffer, as previously described (48). To obtain a single-cell suspension, bacteria were then passed 10 times in and out through a 25G tuberculin syringe. RAW264.7 macrophages were seeded overnight at $5 \times 10^5$ cells/mL in T75 culture flasks and infected at a multiplicity of infection of 10 for 4 hours in T75 culture flasks. Antibiotics were omitted from cell culture medium prior to cell seeding and during the infection. Following a 4-hour infection, 200-µg/mL amikacin was added to the culture flasks for 1 hour to remove extracellular bacteria. Cells were subsequently washed three times, harvested in cell culture media, and seeded on black, clear-bottom 96-well plates at $5 \times 10^4$ cells/well, containing pre-dispensed compound dilutions. Test compounds were prepared as nine twofold serial dilutions in 100% DMSO with a top concentration of 25 µM. BDQ was used as a control compound with a top concentration of 1 µM. High (no growth inhibition) and low (100% growth inhibition) controls were

included in columns 11 and 12, respectively. Medium without cell suspension was added to the low control wells. Final DMSO concentration in test and control wells reached 0.5%. Plates were incubated for 60 hours at 37°C and 5% $CO_2$. Cells were then washed three times with DPBS and lysed in 0.01% Triton X-100 for 10 min. The flash luminescence was recorded upon injecting 1% decanal into each well (25 nL) using Glomax Discover microplate reader. The relative luminescence units were normalized, and the MICs were determined by non-linear regression analysis (GraphPad Prism version 9). The assay was performed in duplicate in two independent experiments.

## Intracellular toxicity

The RAW264.7 cell line was cultured in DMEM supplemented with 10% FBS, 2-mM L-glutamine, 100-U/mL penicillin and 100-μg/mL streptomycin at 37°C and 5% $CO_2$. Cells were seeded on black, clear-bottom 96-well plates at $2.5 \times 10^5$ cells/mL. Test compounds and reference compounds (BDQ and tamoxifen) were added in nine twofold serial dilutions in 100% DMSO with a top concentration of 25 μM. The plate setup included high (no growth inhibition) and low (100% growth inhibition) controls in columns 12 and 1, respectively. Medium without cell suspension was added to the low control wells. Final DMSO concentration reached 1% for test and control wells. Plates were incubated for 60 hours at 37°C and 5% $CO_2$. Before adding the readout reagent, plates were equilibrated at room temperature for 30 min. The CellTiter-Glo version 2.0 Assay (Promega, cat. #G7572) was then added to test wells in a 1:1 ratio and mixed for 2 min on an orbital shaker. The plate was incubated for 10 min at RT to stabilize the luminescent signal. The luminescence was recorded using Glomax Discover microplate reader (Promega). Cell viability was determined by normalization of the recorded signal and establishment of the 50% cytotoxicity concentration based on the non-linear regression curve analysis (GraphPad Prism version 9). The assay was performed in duplicate in two independent experiments.

## ACKNOWLEDGMENTS

The authors thank Stephen Weeks for support in the development of the protein expression platform and Katrien Konings for assistance in the HTS setup.

This work was supported by a grant from the Innovative Health Initiative 2 Joint Undertaking under the RespiriTB (grant no. 853903) project within the IHI AntiMicrobial Resistance Accelerator program. This communication reflects the views of the authors and neither IHI nor the European Union and EFPIA are liable for any use that may be made of the information contained herein.

This work was supported by the Research Fund Flanders (Fonds voor Wetenschappelijk Onderzoek G066619N.

R.G. is a doctoral fellow supported by a DOCPRO4-NIEUWZAP (code 40043) grant awarded to Y.G.-J.S. by the University of Antwerp "Bijzonder Onderzoeksfonds."

Experimental design: N.S., L.O., K.T., D.L., Y.G.-J.S., and D.C.; protein production and purification: N.S., L.O, K.V.C., D.L.S., C.D., and R.G; assay development: H.T.S. and N.S.; cellular assays: N.S. and L.O.; original draft preparation: N.S. and Y.G.-J.S.; review and editing: N.S., L.O., K.V.C., L.D.V., H.P.M., K.A., K.T., S.V.S., Y.G.-J.S, and P.C.; funding acquisition: D.C. and P.C.

## AUTHOR AFFILIATIONS

[1]Laboratory of Microbiology, Parasitology and Hygiene, Faculty of Pharmaceutical, Biomedical and Veterinary Sciences, University of Antwerp, Wilrijk, Antwerp, Belgium
[2]Laboratory of Medical Biochemistry, Faculty of Pharmaceutical, Biomedical and Veterinary Sciences, University of Antwerp, Wilrijk, Antwerp, Belgium
[3]Laboratory of Medicinal Chemistry, Faculty of Pharmaceutical, Biomedical and Veterinary Sciences, University of Antwerp, Wilrijk, Antwerp, Belgium

[4]Biocrystallography, Department of Pharmaceutical and Pharmacological Sciences, KU Leuven, Leuven, Belgium
[5]Janssen Pharmaceutica NV, Beerse, Belgium

## PRESENT ADDRESS

Davie Cappoen, Department of Food, Medicines and Consumer Safety, Scientific Institute of Public Health, Brussels, Belgium

## AUTHOR ORCIDs

Natalia Smiejkowska ⓘ http://orcid.org/0000-0001-7488-6009
Dirk Lamprecht ⓘ http://orcid.org/0000-0003-1066-9026
Yann G.-J. Sterckx ⓘ http://orcid.org/0000-0002-7420-0983
Paul Cos ⓘ http://orcid.org/0000-0003-4361-8911

## FUNDING

| Funder | Grant(s) | Author(s) |
| --- | --- | --- |
| Innovative Health Initiative (IHI) | 853903 | Dirk Lamprecht |
| | | Henri-Philippe Mattelaer |
| | | Davie Cappoen |
| | | Koen Augustyns |
| | | Paul Cos |
| | | Linda De Vooght |
| | | Natalia Smiejkowska |
| | | Lauren Oorts |
| | | Kevin Van Calster |
| Fonds Wetenschappelijk Onderzoek (FWO) | G066619N | Paul Cos |
| | | Lauren Oorts |
| | | Davie Cappoen |
| DOCPRO4-NIEUWZAP Bijzonder onderzoeksfonds University of Antwerp | 40043 | Yann G-J Sterckx |
| | | Rob Geens |

## ADDITIONAL FILES

The following material is available online.

### Supplemental Material

**Figure S1 (Spectrum03723-23-s0001.tif).** Strategy for the recombinant production and purification of Mtr (Mtb) based on an engineered SUMO-fusion construct.
**Captions to supplemental figures, Table S1, supplementary information (Spectrum03723-23-s0002.pdf).** Fig. S1-S3 captions, Table S1, and information concerning synthesis of disulfide mycothiol analogue: BnMS-TNB.
**Figure S2 (Spectrum03723-23-s0003.tif).** Recombinant production and purification of Mtr (Mtb) mutant and Mtr (Mxe).
**Figure S3 (Spectrum03723-23-s0004.tif).** Optimization of GR and Mtr (Mxe) assays.

### Open Peer Review

**PEER REVIEW HISTORY (review-history.pdf).** An accounting of the reviewer comments and feedback.

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
