## [Reviewer comments · Microbiology Spectrum]

Microbiology Spectrum

A high-throughput target-based screening approach for the identification and assessment of *Mycobacterium tuberculosis* mycothione reductase inhibitors

Natalia Smiejkowska, Lauren Oorts, Kevin Van Calster, Linda De Vooght, Rob Geens, Henri-Philippe Mattelaer, Koen Augustyns, Sergei Strelkov, Dirk Lamprecht, Koen Temmerman, Yann Sterckx, Davie Cappoen, and Paul Cos

Corresponding Author(s): Paul Cos, Universiteit Antwerpen

Review Timeline:

Submission Date:	October 20, 2023
Editorial Decision:	November 15, 2023
Revision Received:	December 11, 2023
Accepted:	December 21, 2023

Editor: Antonio Ruzzini

Reviewer(s): The reviewers have opted to remain anonymous.

Transaction Report:

DOI: <https://doi.org/10.1128/spectrum.03723-23>

Re: Spectrum03723-23 (**A high-throughput target-based screening approach for the identification and assessment of Mycobacterium tuberculosis mycothione reductase inhibitors**)

Dear Prof. Paul Cos:

Thank you for submitting your work to Microbiology Spectrum. Your manuscript was reviewed positively by two experts in your field, and appears to be close to publication. Specific reviewer comments and suggestions follow instructions from the Spectrum editorial office.

Revision Guidelines

Sincerely,
Antonio Ruzzini
Editor
Microbiology Spectrum

Reviewer #1 (Comments for the Author):

The authors report a target based high throughput assay development and identification of two novel series of selective Mycobacterium tuberculosis (MTb) mycothione reductase inhibitors. The work described here involved the following:

- a) production of recombinant Mtb mycothione reductase and its Cys39Ser/Cys44Ser double mutant that lacks reductase activity.
- b) Development and miniaturized version of biochemical coupled assays, so that (High throughput screening (HTS) can be performed.

- c) Use of counter screens in the form of substrates, human GR enzyme, to assess selectivity
- d) MIC determination using a Mtb H37Ralux (in which the prokaryotic luciferase luxab genes) strain that enabled the authors to work under L2 conditions.
- e) Next the authors discuss about the inhibitory profile of identified compounds on the recombinant enzyme screen and their MIC50 properties on lab derived Mtb H37Ralux culture, *M. xenopi*.; and murine RAW264.7 macrophage infection model with Mtb H37Ralux.
- f) the authors taking advantage of their developed materials and go on to show two series of inhibitors comprising of Morpholine (Resipiri-1078, Resipiri-1079, Resipiri-1082, Resipiri-1093) and sulfonyl-amine scaffolds that were selective to the MTb mycothione reductase enzyme and exhibited low or weak affinity to human Glutathione reductase.
- g) Further similarity search on a morpholine scaffold in the Janssen library have led to the identification of additional 8 molecules with similar micromolar inhibition at enzyme level, which lacked potency in MIC.

Comments

Overall, the authors have organized and described their work well from achieving the wild and mutant protein production using the recombinant Escherichia coli T7 expression system chaperoned with GroEl-GroES proteins, enzyme characterization using biophysical tools, Assay development and its adaptation to HTS.

- Devising of HTS assay screen with good Z factor and the setting up of important counter assay screens to establish selectivity with different substrate, human glutathione reductase assay, and whole cell assays with Mtb H37Ralux and drug resistant *M. xenopi* strains.
- Emerging SAR from both the series appears to be flat with no major improvement in IC50 as well as MIC50. Another concern is CC50 is almost in the same range as MIC50 seen in macrophages. Of course, these are challenges involved in TB research and might be subject for another research paper.
- The described details are very informative to the researchers in mycobacteria field and could augment the development of selective Mtb Mycothione reductase inhibitor development and the persistent problem that plagues the TB research is in enhancing the MIC in future works.

Minor suggestion

The authors could highlight whether the highlighted 4 inhibitors showed or retained MIC50 activity in the drug resistant *M. xenopi* strain, since drug resistance is a key problem to present TB therapies.

Finally, I suggest the publication of this work with a minor revision.

Reviewer #2 (Comments for the Author):

Smiejkowska et. al. describes the development of an assay followed by a screening campaign of a compound library against Mycobacterium tuberculosis mycothione reductase. The enzyme is key for recycling of Mycothiol a known scavenging antioxidant in Mycobacteria. Selected lead compounds were tested against Mtb in broth and in intracellular model of infection. Overall, this is a strong and solid paper providing the readers with a tailored expression protocol, development of high throughput assay and a good summary of the screening outcome.

The study is well written yet sometimes too elaborative. Some of the beautiful supplementary figures showing the nice expression and purification can be summarized in a more concise way.

Specific remarks:

1. Line 39 lists 19 hits while line 110 and table 1 list 18.
2. Line 77-80 MSH is only an antioxidant. It also reacts with Nitric oxide and other nitrogen species (PMID: 17638697) and PMID: 37798648 for MSH role in signaling.
3. Figure 5 describes well the screening, validation assays and the exact MIC details both intracellularly and in vitro are well presented in Table 1. Yet the issue of specificity needs to be better elaborated. I suggest, if possible, assaying the spectrum activity of selected compounds against a panel of representative microbes such as gram positive, gram negatives and mycobacteria. One would expect that the activity would be limited to actinomycetes and if not, there might be a polar activity to discuss.

Title: A high-throughput target-based screening approach for the identification and assessment of *Mycobacterium tuberculosis* mycothione reductase inhibitors

Authors: Natalia Smiejkowska, Lauren Oorts, Kevin Van Calster, Linda De Vooght, Rob Geens, Henri-Philippe Mattelaer, Koen Augustyns, Sergei V. Strelkov, Dirk Lamprecht, Koen Temmerman, Yann G.-J. Sterck, Davie Cappoen, Paul Cos

The authors report a target based high throughput assay development and identification of two novel series of selective *Mycobacterium tuberculosis* (*MTb*) mycothione reductase inhibitors. The work described here involved the following:

- a) production of recombinant *Mtb* mycothione reductase and its Cys39Ser/Cys44Ser double mutant that lacks reductase activity.
- b) Development and miniaturized version of biochemical coupled assays, so that (High throughput screening (HTS) can be performed.
- c) Use of counter screens in the form of substrates, human GR enzyme, to assess selectivity
- d) MIC determination using a *Mtb* H37Ra*lux* (in which the prokaryotic luciferase *luxab* genes) strain that enabled the authors to work under L2 conditions.
- e) Next the authors discuss about the inhibitory profile of identified compounds on the recombinant enzyme screen and their MIC₅₀ properties on lab derived *Mtb* H37Ra*lux* culture, *M. xenopi.*; and murine RAW264.7 macrophage infection model with *Mtb* H37Ra*lux*.
- f) the authors taking advantage of their developed materials and go on to show two series of inhibitors comprising of Morpholine (Resipiri-1078, Resipiri-1079, Resipiri-1082, Resipiri-1093) and sulfonyl-amine scaffolds that were selective to the *MTb* mycothione reductase enzyme and exhibited low or weak affinity to human Glutathione reductase.
- g) Further similarity search on a morpholine scaffold in the Janssen library have led to the identification of additional 8 molecules with similar micromolar inhibition at enzyme level, which lacked potency in MIC.

Comments

Overall, the authors have organized and described their work well from achieving the wild and mutant protein production using the recombinant *Escherichia coli* T7 expression system chaperoned with GroEl-GroES proteins, enzyme characterization using biophysical tools, Assay development and its adaptation to HTS.

- Devising of HTS assay screen with good Z factor and the setting up of important counter assay screens to establish selectivity with different substrate, human glutathione reductase assay, and whole cell assays with *Mtb* H37Ralux and drug resistant *M. xenopi* strains.
- Emerging SAR from both the series appears to be flat with no major improvement in IC₅₀ as well as MIC₅₀. Another concern is CC₅₀ is almost in the same range as MIC₅₀ seen in macrophages. Of course, these are challenges involved in TB research and might be subject for another research paper.
- The described details are very informative to the researchers in mycobacteria field and could augment the development of selective *Mtb* Mycothione reductase inhibitor development and the persistent problem that plagues the TB research is in enhancing the MIC in future works.

Minor suggestion

The authors could highlight whether the highlighted 4 inhibitors showed or retained MIC₅₀ activity in the drug resistant *M. xenopi* strain, since drug resistance is a key problem to present TB therapies.

Finally, I suggest the publication of this work with a minor revision.

Reply to the referees (Spectrum03723-23)

A high-throughput target-based screening approach for the identification and assessment of *Mycobacterium tuberculosis* mycothione reductase inhibitors

Natalia Smiejkowska, Lauren Oorts, Kevin Van Calster, Linda De Vooght, Rob Geens, Henri-Philippe Mattelaer, Koen Augustyns, Sergei Strelkov, Dirk Lamprecht, Koen Temmerman, Yann Sterckx, Davie Cappoen, Paul Cos

First and foremost, we would like to thank **reviewer #1** for a thorough assessment of the submitted manuscript and for providing constructive suggestions.

Comment 1: 'The authors could highlight whether the highlighted 4 inhibitors showed or retained MIC50 activity in the drug resistant *M. xenopi* strain, since drug resistance is a key problem to present TB therapies.'

We thank the reviewer for this suggestion and have performed additional experiment to follow up on this matter. We tested the three hit compounds (Respiri-1079, -1082, 1093) against *M. xenopi* wild-type to assess whether we could observe a lack of inhibitory activity against Mtr_{Mxe} as seen during the in vitro enzyme activity assays. The results are shown below for the reviewer's and editor's convenience, but we propose not to include these data in the manuscript. The results demonstrate a bactericidal effect against *M. xenopi* wild-type, which we attribute to the general cytotoxic, low-potency profile of these compounds (as observed in the RAW macrophage cytotoxicity assay). We did not perform the experiments with a drug-resistant *M. xenopi* strain as it is not available in-house. However, the lack of compound activity in the drug-susceptible strain suggests the observed effect will be similar in the drug-resistant strain.

We also thank **reviewer #2** for a thorough assessment of the submitted manuscript and for providing constructive suggestions.

Comment 1: 'The study is well written yet sometimes too elaborative. Some of the beautiful supplementary figures showing the nice expression and purification can be summarized in a more concise way.'

We have shortened the figure legend of Figure S1.

Figure S1. Strategy for the recombinant production and purification of Mtr_{Mtb} based on an engineered SUMO-fusion construct. (A) Schematic overview of the SUMO-Mtr_{Mtb} fusion construct. **(B)** Recombinant production of SUMO-Mtr_{Mtb} in *E. coli* (top panel). The culture samples were analyzed by SDS-PAGE and anti-SUMO Western blot (WB): before addition of arabinose for chaperone production (lane 1), before addition of IPTG for target protein production (lane 2), 16 h post-IPTG induction (lane 3) and sonicated lysate (lane 4), Protein Molecular Weight Marker (lane M). The IEX elution profile (black trace) consists of two peaks eluting at 30 mM and 200 mM (NH₄)₂SO₄ (indicated by the magenta and orange diamonds, respectively). SDS-PAGE analysis shows that the first and second peak contain the GroEL chaperone (MM ~58 kDa) and SUMO-Mtr_{Mtb} (MM ~61 kDa), respectively. The fractions under the second peak are pooled and subjected to SUMO-tag cleavage, followed by SDS-PAGE analysis. The corresponding bands were excised and sent for LC-MS analysis to confirm the identities of the GroEL chaperone and SUMO-cleaved Mtr_{Mtb}: IEX peak 1 (lane 1), IEX peak 1 treated with SUMO protease (lane 2), IEX peak 2 (lane 3), IEX peak 2 treated with SUMO protease (lane 4, MM Mtr_{Mtb} ~50 kDa), Protein Molecular Weight

Marker (lane M). **(C)** SUMO-protease treatment of SUMO-Mtr_{Mtb} is followed by its purification by tandem IEX and SEC. SDS-PAGE (left) and anti-SUMO Western blot (right) analysis indicate the successful cleavage of the SUMO-tag and purification of Mtr_{Mtb}: IEX peak 2 (lane 1), IEX peak 2 treated with SUMO-protease (lane 3), tandem IEX anion step elution peak (lane 3), SEC peak (lane 3), Protein Molecular Weight Marker (lane M)."

Comment 2: 'Line 39 lists 19 hits while line 110 and table 1 list 18'

We have amended this. Line 110 indeed mentioned 18 compounds, while there are 19 in total. Table 1 lists 19 compounds according to us.

Comment 3: 'Line 77-80 MSH is only an antioxidant. It also reacts with Nitric oxide and other nitrogen species (PMID: 17638697) and PMID: 37798648 for MSH role in signaling.'

We agree with reviewer #2 and have included the additional references in the introduction section as follows:

"A significant amount of interest has been directed towards mycothiol (MSH), a low molecular weight thiol that plays a role in maintaining a reducing environment as a key antioxidant^{9,10}, in RNS protection¹¹, and NO signaling (the latter was documented in *Streptomyces* spp.)¹²."

Comment 4: 'Figure 5 describes well the screening, validation assays and the exact MIC details both intracellularly and in vitro are well presented in Table 1. Yet the issue of specificity needs to be better elaborated. I suggest, if possible, assaying the spectrum activity of selected compounds against a panel of representative microbes such as gram positive, gram negatives and mycobacteria. One would expect that the activity would be limited to actinomycetes and if not, there might be a polar activity to discuss'

We understand the reviewer's point of view. However, as outlined in the manuscript the focus of the IMI consortium lies on mycobacteria, which is why we are of the opinion that screening against other bacterial pathogens is beyond the scope of the current manuscript. In our view, further exploration of the identified compound series would only be possible after SAR optimization with the specific goal of improving compound potency (and lowering general, non-specific cytotoxicity).

Re: Spectrum03723-23R1 (**A high-throughput target-based screening approach for the identification and assessment of *Mycobacterium tuberculosis* mycothione reductase inhibitors**)

Dear Prof. Paul Cos:

Thank you for taking the time to respond to the reviewer comments and suggestions. I am satisfied with the updated manuscript and your responses, and I look forward to reading your follow up studies.

Your manuscript has been accepted, and I am forwarding it to the ASM production staff for publication. Your paper will first be checked to make sure all elements meet the technical requirements. ASM staff will contact you if anything needs to be revised before copyediting and production can begin. Otherwise, you will be notified when your proofs are ready to be viewed.

Sincerely,
Antonio Ruzzini
Editor
Microbiology Spectrum